# Non-Human Primate-Derived Adenoviruses for Future Use as Oncolytic Agents?

**DOI:** 10.3390/ijms21144821

**Published:** 2020-07-08

**Authors:** Selas T.F. Bots, Rob C. Hoeben

**Affiliations:** Department of Cell and Chemical Biology, Leiden University Medical Center, 2333 ZC Leiden, The Netherlands; r.c.hoeben@lumc.nl

**Keywords:** non-human primate adenovirus, human adenovirus, oncolytic virus, adenoviral vector, taxonomy, genetic recombination, oncolysis, anti-tumor immunity

## Abstract

Non-human primate (NHP)-derived adenoviruses have formed a valuable alternative for the use of human adenoviruses in vaccine development and gene therapy strategies by virtue of the low seroprevalence of neutralizing immunity in the human population. The more recent use of several human adenoviruses as oncolytic agents has exhibited excellent safety profiles and firm evidence of clinical efficacy. This proffers the question whether NHP-derived adenoviruses could also be employed for viral oncolysis in human patients. While vaccine vectors are conventionally made as replication-defective vectors, in oncolytic applications replication-competent viruses are used. The data on NHP-derived adenoviral vectors obtained from vaccination studies can only partially support the suitability of NHP-derived adenoviruses for use in oncolytic virus therapy. In addition, the use of NHP-derived adenoviruses in humans might be received warily given the recent zoonotic infections with influenza viruses and coronaviruses. In this review, we discuss the similarities and differences between human- and NHP-derived adenoviruses in view of their use as oncolytic agents. These include their genome organization, receptor use, replication and cell lysis, modulation of the host’s immune responses, as well as their pathogenicity in humans. Together, the data should facilitate a rational and data-supported decision on the suitability of NHP-derived adenoviruses for prospective use in oncolytic virus therapy.

## 1. Introduction

Adenoviruses have long been used as a probe to study cellular and viral processes as well as a tool for gene transfer and gene therapy. They were first identified by Rowe and colleagues in 1953 as a transmissible agent responsible for the degeneration of cell lines established from adenoidal tissue and tonsils [1]. These so-called “adenoviruses” attracted particular attention after the discovery that certain isolates (i.e., species-A types *Human Adenovirus* (HAdV)-A12 and HAdV-A18, see taxonomic notes below) can induce tumors in newborn hamsters, whereas others (i.e., species-C types HAdV-C2 and HAdV-C5) are not or only very weakly oncogenic [2,3]. As such, adenoviruses have played an important role in biomedical research. They provided model systems to study the organization of eukaryotic genes and the regulation of their expression, as well as to study the mechanisms of deoxyribonucleic acid (DNA) replication in mammalian cells (as reviewed in [4,5]). Such studies not only led to detailed insight in many cellular and viral processes, but also yielded various broadly applicable tools and techniques, such as adenoviral gene transfer vectors and oncolytic viruses.

Adenoviruses are non-enveloped viruses with an icosahedral capsid and a large double-stranded DNA genome of roughly 34–36 kB in size. The family of *Adenoviridae* consists of five genera, of which the genus *Mastadenovirus* includes all adenoviruses that infect primates. Adenoviruses are conventionally named based on their host species (Table 1). This is informative since adenoviruses have a narrow host range and primarily infect one species only. The adenoviruses isolated from humans and remaining hominids also have a narrow host range, albeit that under experimental conditions, these viruses can infect several non-human primate (NHP) species. The human adenoviruses are classified into seven “species” (formerly called “subgroups”) named *Human Adenovirus* (HAdV-) *A* through *G* [6]. Improved detection and sequencing techniques have resulted in a massive expansion of the number of human adenoviruses. The Human Adenovirus Working Group has recorded up to 103 types as of last year (http://hadvwg.gmu.edu/). The classification of the adenovirus types used to be based on serology, hemagglutination, and restriction analyses, but more recently includes whole genome sequencing, with the focus on sequence similarity, genome organization, and GC content [7,8]. Based on these criteria, several NHP-derived adenoviruses have been classified into the HAdV species (Figure 1). In fact, all HAdV species, with the exception of HAdV-D, encompass both human- and NHP-derived adenoviruses although they do have a predominant host species [9,10,11]. Contrarily, there are several *Simian Adenovirus* (SAdV) species for which there are no known human isolates. Research into the ancestry of the HAdV has estimated that most species originate from NHPs and have switched host over the course of evolution [11,12]. However, the frequency by which cross-species transmission of adenoviruses occurs is thought to be very low [12]. In support of this, there are hitherto no reports of humans infected with viruses highly similar or identical to NHP-derived adenoviruses.

In contrast, human adenoviruses circulate profusely in the human population. The prevalence of human adenoviruses is dependent on geographic location and sample collection, as they associate with different clinical manifestations. Members of HAdV-A, HAdV-B, and HAdV-C are most prevalent as illustrated by the presence of neutralizing antibodies (NAbs) in 60–80% of the population [13]. In immune-competent individuals, human adenoviruses cause only a mild and self-limited disease. The high safety profile in combination with their broad immunogenicity has made several human adenoviruses popular as vectors for vaccine development and gene therapy strategies. However, pre-existing immunity may limit the efficacy of some adenoviral vector applications as they are swiftly cleared from the blood by NAbs upon intravenous administration. This route of administration would be favorable, as opposed to intratumoral injection, as it is least invasive. In oncolytic applications especially, secondary tumors can have gone undetected and, more importantly, not all tumors are accessible by needle. To date, most adenoviral vectors are based on the species C *Human Adenovirus* type 5 (HAdV-C5), globally one of most prevalent adenoviruses [13,14]. Consequently, clinical trials using HAdV-C5-based vectors have demonstrated moderate and variable results, in contrast to pre-clinical studies where naïve animal models are used. As a means to circumvent pre-existing immunity, research has gained interest in the use of human adenovirus types with low seroprevalence in the population. The hunt for “rare serotypes” has resulted in the use of human adenoviruses from alternative species, i.e., HAdV-D26, HAdV-B35, HAdV-D48, and HAdV-D64. However, the ability of these adenoviral vaccine vectors to induce an immune response demonstrated to be not as potent as HAdV-C5 [15]. Therefore, the discovery of new adenovirus types has become of major interest over the last decade. Due to their shared origin with human adenoviruses, NHP-derived adenoviruses have been employed to create new adenoviral vectors for vaccine development [16]. So far, there is no evidence of NHP-derived types circulating in the human population. As a result, the presence of NAbs against NHP-derived adenoviruses is usually lower than against human types [17]. The low seroprevalence of neutralizing immunity against NHP adenoviruses may reduce the variations in clinical responses observed in many studies, and may lead to an increase in overall efficacy. Therefore, NHP-derived adenoviruses might also provide a valuable alternative as oncolytic agents with limited neutralizing immunity in the human population.

Oncolytic viruses are viruses that preferentially infect, replicate in, and kill cancer cells as opposed to healthy cells. Again, human adenoviruses are one of the leading candidate viruses due to their high immunogenicity and their safety profile. Moreover, their long history as viral vectors has generated the means to utilize oncolytic adenoviral vectors as carriers of, i.e., immune-modulating transgenes and tumor antigens, thereby acting as an oncolytic virus and a cancer vaccine simultaneously [18]. So far, the use of NHP-derived adenoviruses as oncolytic agents has not been studied and has mostly been restricted to non-replicating vectors for vaccine development (Table 2). Unfortunately, data regarding replicating-incompetent vectors cannot be directly extrapolated to replicating-competent oncolytic viruses. In addition, oncolytic viruses sometimes require different characteristics (i.e., tropism and immune responses) or additional traits (i.e., replication and cell killing) compared to vaccine vectors. Moreover, the recently increased awareness for zoonoses highlights the concern anent safe administration of non-human viruses to cancer patients and the pathologies that could arise from it. The aim of this review is to address the aforementioned issues and to come to a rational and data-supported decision on the use of NHP-derived adenoviruses as a base for the development of oncolytic adenoviruses. To this end, the differences and similarities between human- and NHP-derived adenoviruses regarding their use as oncolytic agents will be discussed.

## 2. Genome Organization and Recombination

The continued discovery of new adenoviruses and their genome sequences confirm the similarity between human- and NHP-derived adenoviruses. While phylogenetic analyses of complete genomes are desirable, nucleotide sequencing of *DNA polymerase* (*pol*), *hexon*, and *penton-base* genes can discriminate nearly all adenovirus types [19]. The structure of phylogenetic trees generated from the sequences of individual genes can sometimes vary for the same panel of adenoviruses. This gene-dependent topology is the result of genetic recombination, one of the key characteristics of adenoviruses [6]. Recombination can involve part of genes, complete genes, or entire gene regions [20], and does not seem to be inherent to human-derived adenoviruses only. In NHPs, co-infection of adenoviruses belonging to the same adenovirus species or even different species was observed, ergo opening the possibility for genetic recombination [12]. Indeed, discordant structures in the relationship between NHP-derived adenoviruses were observed upon severed analysis of *pol*, *hexon*, and *penton-base* gene sequences [21]. Interestingly, recombination of adenoviruses is not limited to adenoviruses derived from the same species. The topology of phylogenetic trees with the *hexon* and *fiber* gene sequences of human and simian-derived adenoviruses demonstrated the likeliness of intra-species recombination [22]. This was later evidenced by the *fiber* region of HAdV-B3, which was shown to be most closely related to the *fiber* of simian B species SAdV-B32, while other genomic regions had the highest sequence identity with HAdV-B7 [23]. Hence, the authors suggested that HAdV-B3 had arisen from a recombination between HAdV-B7 and SAdV-B32. In addition, dissimilar relationships between human- and NHP-derived HAdV-B viruses concerning the major capsid proteins (hexon, fiber, and penton-base) as well as pVII-pVI demonstrated extensive recombination events in the past between these viruses [24]. Recently, a human-derived adenovirus (HAdV-B76) was shown to be the derivative of a recombination between three adenoviruses that originated from human, chimpanzee, and bonobo [20]. The discovery of HAdV-B76 illustrates the unrestrained breadth of intra-species transmission of adenoviruses, which, according to the authors, resembles a game of evolutionary “ping-pong” between multiple primate species. Moreover, HAdV-B76 demonstrated high similarity with the major capsid proteins of HAdV-B16 (fiber) and HAdV-B21 (hexon and penton-base) [20]. The authors further hypothesized that the restricted sequencing of specific viral genes could have resulted in the misclassification of some adenovirus types. Together, these findings underscore the importance of the decision to include whole genome sequencing in the classification of adenovirus types [7]. In summary, it appears that recombination is a characteristic of both human- and NHP-derived adenoviruses, and spans the hominid species-barriers.

It is not surprising that recombination is most often observed for the major capsid proteins, as these are under direct immunological pressure. Nevertheless, recombination does not seem to appear to be restricted to these regions. For HAdV-C, no recombination could be detected for the minor capsid proteins (IIIa, V-IX), yet all early (*E*) gene regions demonstrated multiple recombination events [25]. More importantly, these regions could be exchanged independent of one another, regardless of their intergenic relations. We and others have also observed the *E3* region as a highly variable region with sequence identity percentages well below those observed for the major capsid proteins [22,24]. This is especially interesting, as the *E3* region is involved in the modulation of the host’s cellular immune responses. The size and complexity of the *E3* region varies between HAdV species. The variation is reflected in the different affinities for cellular receptors of the same family, as illustrated by the use of a variety of the signaling lymphocyte activation molecule (SLAM) receptors by the E3 conserved region 1 (CR1) proteins (CR1-α, CR1-β, and CR1-γ) of multiple adenovirus types [26]. However, as the receptors belong to the same family, the immunomodulatory effects might be similar. Although no studies have been performed regarding the function of the *E3* genes in NHP-derived adenoviruses, the E3 open reading frames (ORF) were shown to be conserved between the human- and NHP-derived adenoviruses [27]. However, substantial differences were observed for the ORFs of CR1-α, CR1-β, and CR1-γ. For human-derived adenoviruses, the CR1-proteins were shown to diminish immune cell adhesion, modulate T cell receptor signaling, and inhibit natural killer (NK) cell-mediated killing [26]. Hence, it would be interesting to determine whether the *E3 CR1* genes of NHP-derived adenoviruses have similar affinities for the SLAM receptor family. The implications of the diversity of the *E3* region between (non-)human adenovirus types has been under-investigated in vivo, as the *E3* region is routinely deleted for vector design in vaccination and gene therapy studies. However, with the use of replication-competent virus for oncolytic virotherapy, the immunomodulatory effects of E3 proteins could be desirable. For HAdV-C5, its oncolytic potency was much reduced in vivo upon deletion of the *E3* region [28]. The use of an athymic mouse model in this study suggests that the majority of the *E3* genes are probably involved in the evasion of NK cells, a subset that is also important in anti-tumor immunity. Interestingly, human and NHPs develop contrasting humoral and cellular immune responses upon adenoviral infection [27]. This could possibly have led to the adaptation of E3 to its host immune response and might, as a consequence, be less effective in another species. Therefore, the characterization of the *E3* region of (non-)human primate adenoviruses would give a rational for the decision to retain, mutate, or delete (part of) the *E3* region in an adenoviral oncolytic agent. In conclusion, similar genetic variability is observed among human adenoviruses as well as between human- and NHP-derived adenoviruses regarding the major capsid proteins as well as the *E3* region (Figure 2). While the implications of the variability observed in *E3* for oncolytic virotherapy remain to be established, there appear to be no distinguishable genomic features in human or NHP-derived adenoviruses that would make either virus preferable as oncolytic agent.

## 3. Receptor Use

In most of the *Mastadenoviruses*, adenovirus entry is mediated via attachment of sequences in the fiber knob protein to the primary attachment receptor, followed by binding of the Arg-Gly-Asp (RGD) motif in the penton-base protein to integrins on the cell surface. This triggers internalization of the virion by endosomal uptake. Adenoviruses make use of a variety of cell receptors for primary attachment, as reviewed in Arnberg et al. [32]. The coxsackie and adenovirus receptor (CAR) is often the primary receptor of human-derived adenoviruses, yet additional receptors like CD46, CD80/CD86, Desmoglein-2, and sialic acids are also being exploited [32,33]. The use of receptors other than CAR could be advantageous for oncolytic virotherapy as CAR is not ubiquitously expressed on cells and its expression is often downregulated in tumors [34,35]. For most HAdV species, all types generally use the same entry receptor. Therefore, it was anticipated that chimpanzee adenovirus type 68 (ChAd68), classified as HAdV species E, uses CAR as its primary entry receptor [36]. Likewise, the primary receptor for SAdV-21, a HAdV-B virus otherwise referred to as AdC1, was shown to bind to CD46 [37]. Moreover, several rhesus monkey-derived adenoviral vectors were shown to use CAR as their primary receptor for entry into human cell lines [38]. It is noteworthy that adenoviruses derived from new world monkeys (NWM), which are more distantly related to humans compared to the great apes, share the same attachment receptor(s). This might imply that the use of entry receptors has been conserved over a long period of time, and speaks in favor of similar entry strategies for both human- and NHP-derived adenoviruses. However, in the same study by Abbink et al. none of the rhesus-derived adenoviruses out a panel of 15 viruses used CD46 as a receptor [38]. Taken together, human- and NHP-derived adenoviruses share several attachment receptors, but not all receptors might be conserved between these relatively similar adenoviruses derived from closely related species.

The assumption that adenoviruses within one adenovirus species make use of the same receptor should be taken with caution, as the attachment receptor(s) of many human- and NHP-derived adenoviruses have not been fully delineated. This has been illustrated clearly for a number of human-derived adenovirus B types, which use a diversity of receptors (CD46, CD80/CD86, Desmoglein-2, heparan sulfate proteoglycans) and some of the types have the capacity to bind multiple of these receptors. Other species also exhibit this receptor heterogeneity. For example, two human-derived adenoviruses from separate species, HAdV-D26 and HAdV-G52, were only recently shown to interact with CAR as well as with sialic acids [39,40]. More importantly, sialic acids appeared to be the primary attachment receptor for both viruses. For oncolytic virotherapy, the use of sialic acids as a primary receptor is favorable as they are ubiquitously expressed on cells, thus broadening the tropism of the vector. Several NHP-derived adenoviruses can also make use of multiple receptors for entry into a cell. An *E1/E3*-deleted adenoviral vector that originated from a chimpanzee-derived isolate Y25 and is classified as HAdV-E, named ChAdOx1, transduced CHO cells independent of CAR or CD46 expression, although transduction of CHO-CAR cells was slightly more efficient [41]. In addition, blocking of CAR with HAdV-C5 fiber knob protein did not alter the transduction of A549 cells. Therefore, ChAdOx1 seemingly uses additional receptors for entry. HAdV-G52 is closely related to the remaining NHP-derived HAdV-G adenoviruses, hence its use of sialic acids for cell entry might be applicable to NHP-derived adenoviruses as well. Therefore, sialic acids could make a candidate receptor for ChAdOx1. On the other hand, in the panel of isolates described by Abbink et al., one-third of the rhesus-derived adenoviruses transduced cells regardless of CAR or CD46 expression, and the knock-down of sialic acids had only minor to moderate effects on transduction efficacy [38]. Therefore, the authors suggested that these viruses exploit an alternative yet unidentified receptor for entry into the host cell. To date, the identification of primary attachment receptors for NHP-derived adenoviruses is lagging behind. In addition, while there is a detailed characterization of the viral interactions with CAR and CD46, the interaction with other known receptors has received little attention [42]. The incomplete characterization of NHP-derived adenovirus receptors leaves their tropism as of yet unexplored. This seems counterintuitive, considering that there is great clinical relevance of the attachment receptor. Evidently, a lack of receptor expression forms a direct impasse for cell entry and subsequent tumor cell killing in oncolytic virus applications. Therefore, insight in receptor specificity by NHP-derived adenoviruses is necessary to ensure efficacy upon their use in oncolytic virotherapy.

## 4. Replication and Cell Lysis

The use of the appropriate receptors does not always warrant for effective replication. In some cases, this is attributable to the host species-barrier, which is exemplified by the use of human-derived adenoviruses in mouse models. Human-derived adenoviruses generally do not replicate in murine cell lines. However, some murine cell lines are semi-permissive and support the replication of oncolytic vectors based on HAdV-C5 [43]. Nevertheless, the permissibility of these murine cell lines is adenovirus species-dependent. A mouse epithelial cell line (NMuMG) was shown to support replication of a series of HAdV-C, HAdV-D, and HAdV-E human-derived adenovirus types but not of HAdV-A, HAdV-B, and HAdV-F types [44]. Moreover, lentiviral transduction of NMuMG with CD46 did not rescue the replication of a chimeric HAdV-B *E3/E4ORF4*-deleted oncolytic virus named EnAd, previously known as ColoAd1 [45]. Taken together, the replication of human-derived adenovirus types seems to be affected to varying degrees by intrinsic cellular differences between murine and human cells.

These intrinsic cellular differences might also be reflected in the replication potential of NHP-derived adenoviruses in human cells. As discussed previously, all of the adenovirus *E* gene regions involved in viral replication (*E1, E2*, and *E4*) exhibit some variability between different human-derived adenovirus strains [25]. Whole genome sequencing of NHP-derived adenoviruses also revealed moderate diversity within the *E1* region compared to human-derived adenoviruses (Figure 2C). It could be hypothesized that the genomic variability causes differences in affinity for cellular co-factors, as observed for homologous *E3* genes of different adenovirus types [26]. These slight variations in receptor affinity could hamper effective replication of NHP-derived adenoviruses in human cells. Contrarily, we observed variable but overall strong oncolytic effects of several NHP-derived adenovirus isolates in a large panel of human cancer cell lines [46]. As the susceptibility of human cancer cell lines to the NHP-derived adenoviruses was type-dependent, this could still support the hypothesis that genomic variation leads to slight variations in co-factor affinities that govern the variability in replication capacity and oncolytic abilities. However, little is known about the steps following the internalization of human- and NHP-derived adenoviruses, especially. Furthermore, there have been no additional studies parallel-testing the replication and oncolytic potential of human- and NHP-derived oncolytic adenoviruses in human cancer cell lines or more complex pre-clinical models. Therefore, it remains to be seen whether the oncolytic potential of NHP-derived adenoviruses is in fact limited to a few types. For example, an *E3*-deleted oncolytic adenoviral vector based on chimpanzee-derived adenovirus AdC7 with *E1A* expression under a survivin promoter was shown to replicate in only a limited number of human cancer cell lines and with moderate cytotoxic effects, even at a high multiplicity of infection (MOI) [47]. In conclusion, it seems that the inability of some NHP-derived adenoviruses to efficiently replicate in and lyse human tumor cells is determined by adenovirus type rather than intra-species variation, although the latter might underlie some of the observed differences. Research into the cellular interactions that mediate the particular stages of viral replication, and the variations between types therein, could provide insights into the diversity that is observed with regard to viral replication and cell killing.

The induction of an immunogenic cell death (ICD) is crucial in oncolytic virotherapy, as it can surmount the immunosuppressive tumor microenvironment. Interestingly, adenovirus types can induce different forms of cell death. For example, EnAd was shown to mediate a form of ICD that was substantially different from, and more immunogenic than, HAdV-C5-induced cell death [48]. To our knowledge, no studies have looked into characteristics of virus-induced cell death by NHP-derived adenoviruses. However, we repeatedly observed faster killing of human cancer cell lines by HAdV species B NHP-derived adenoviruses compared to species C NHP-derived adenoviruses, suggestive of the use of alternative cell death pathways [own observations]. Therefore, it would be interesting to verify these differences between various adenovirus types, whether they are adenovirus species-specific or host species-specific, and to correlate them to subsequent anti-tumor immune responses.

## 5. Immune Responses

The oncolytic potency of viruses is not solely determined by their capacity to induce tumor cell death, but rather relies on the combination of their oncolytic potential and their ability to elicit an immune response directed against the tumor cells. Although there are no data on the ability of NHP-derived oncolytic adenoviruses to induce immune responses, the responses to NHP-derived adenoviral vaccine vectors and/or gene delivery vectors have been studied in depth. For vaccine development, adenoviruses are routinely made into replication-incompetent adenoviral vectors by deletion of (part of) *E1*, and in some cases additional deletion(s) of *E3* and/or *E4*. Although this presumably influences their ability to modulate immune responses, it can be assumed that the magnitude rather than the phenotype of the immune response would differ from replication-competent oncolytic vectors.

Antigen-presenting cells (APCs) such as macrophages and dendritic cells (DCs) are key regulators in the induction of an immune response. The recruitment of DCs is dependent on the location and the vector used. Replication-competent HAdV-C5 was shown to recruit and activate several DC subsets in the circulation and the periphery, including plasmacytoid and myeloid DCs as well as Langerhans in the mucosa [49]. In the draining lymph nodes (dLNs), the relative distribution of the recruited DC subsets was comparable between an *E1/E3/E4*-deleted HAdV-C5 vector and several NHP-derived *E1*-deleted adenoviral vectors, although the total number of DCs was variable [50]. In response to an adenoviral vector, DCs migrate to the dLNs and stimulate proliferation of naïve T cells via cross-presentation of antigens, resulting in the induction of an antigen-specific T cell response [49,50]. Both human- and NHP-derived adenoviral vectors induce CD8^+^ T cell as well as CD4^+^ T cell responses [17,51]. In anti-tumor immunity, the main effector cells are CD8^+^ T cells. For human-derived adenoviral vectors, there is great diversity in their potency to induce such CD8^+^ T cell responses. Interestingly, a similar variability is seen for NHP-derived adenoviral vectors [17]. Moreover, the magnitude of the response appeared to be independent of host origin when human- and NHP-derived adenoviral vectors were tested in parallel. This is further supported by the similar hierarchy of the adenovirus species regarding their ability to induce T cell responses. Adenoviral vectors of HAdV-C, derived either from humans or from NHPs, induce the strongest cellular immune response compared to adenoviruses of any of the other species [52]. Regardless, most adenoviral vectors, being either human or NHP-derived, are considerably less potent. As a result, vectors derived from HAdV-C5 are often superior to other human- as well as NHP-derived adenoviral vectors [51]. Nevertheless, *E1*-deleted chimpanzee adenovirus type 3 (ChAd3) and three *E1*-deleted gorilla-derived adenoviral vectors (GC44, GC45, and GC46) were shown to induce cellular immune responses which were comparable to replication-incompetent HAdV-C5 in mice [17,53]. Not surprisingly, these vectors all belong to HAdV-C. Collectively, these data indicate that there is the potential to find an NHP-derived adenoviral vector that would be equivalent to HAdV-C5.

One should be mindful of the interpretation of the cellular immune response to adenoviral vectors as it is reliant on multiple factors. Firstly, the immune responses induced in mice upon vaccination with chimpanzee-derived adenoviral vectors were shown to be not only vector-dependent, but also mouse strain-dependent, presumably due to variations in haplotype and innate immunity [54]. In addition, all mouse models are inherently naïve to human-derived adenoviruses and therefore do not consider the influence of pre-existing immunity. This could be of lesser concern for NHP-derived adenoviruses due to their low seroprevalence, and might consequently influence the interpretation of the data. On the other hand, pre-existing immunity to human-derived adenoviruses can still result in the neutralization of NHP-derived adenoviruses due to cross-neutralization [55]. However, two chimpanzee-derived adenoviral vectors (ChAd3 and PanAd3) induced potent and comparable immune responses in mice, macaques, and humans [52]. Therefore, the impact of cross-neutralization might either be limited or type-dependent. Secondly, the magnitude of the cellular immune response is both adenovirus type-dependent as well as dose-dependent. Several studies have shown that an increasing viral dose does not necessarily lead to stronger T cell responses in case of potent adenoviral vectors [51,53,56]. This suggests that these potent vectors reach a “plateau” after which higher amounts of virus do not lead to improved responses. In contrast, less potent human- and NHP-derived adenoviral vectors induce increasingly stronger T cell responses at higher doses and responses become rapidly weaker and/or delayed upon lower dosage [17,51,57]. Therefore, optimization of the viral dose is key when comparing different adenoviral vectors.

Dose-adjusted comparisons of human- and NHP-derived adenoviral vectors have yielded more insight in the viral cultivation of the immune landscape and the differences therein. CD8^+^ T cells can be subdivided into four subsets based on surface expression markers. These subsets include naïve cells (T_n_), short-lived effector cells (T_eff_), and longer-lived memory CD8^+^ T cells, the latter of which can be further subdivided into effector memory T cells (T_em_) and central memory T cells (T_cm_), and are reviewed elsewhere [58]. The proportionate importance of the T_eff_, T_em_, and T_cm_ subsets has gained attention in the field of immunotherapy recently, as cytotoxicity along with memory formation seem essential for durable anti-tumor immunity. In humans, vaccination with a ChAd3 vector was shown to induce all of the aforementioned T cell subsets while the number of regulatory T cells remained unchanged [59]. The functionality of T cells is defined by their polyfunctionality, often measured in their ability to produce interferon (IFN) γ, tumor necrosis factor (TNF) α, and/or interleukin (IL)-2 simultaneously. Although a HAdV-C5 vector induces higher frequencies of total T_eff_ cells compared to other (non-)human primate-derived adenoviral vectors, the NHP-derived vectors were shown to induce comparable or higher frequencies of polyfunctional (IFNγ^+^TNFα^+^IL-2^+^) CD8^+^ T cells at similar doses [17]. As expected, lowering the dose of HAdV-C5 resulted in a relatively higher amount of polyfunctional CD8^+^ T cells. However, this increase was solely due to a decrease in the frequency of T_eff_ cells producing either IFNγ and TNFα (+2), or IFNγ alone (+1), and not an increase in polyfunctional (+3) T_eff_ cells. Therefore, NHP-derived adenoviruses might compensate with a higher quality of the immune response rather than quantity (magnitude), as seen for HAdV-C5. Furthermore, the observed reduced polyfunctionality for HAdV-C5 could indicate a malfunctioning of the CD8^+^ T cell response. This could be either due to partial activation of the T_eff_ cells or due to functional exhaustion of the immune cells. In agreement with the latter, vaccination with *E1/E3*-deleted HAdV-C5 resulted in higher PD-1 expression on antigen-specific CD8^+^ T cells compared to the use of low-prevalent human adenoviruses [57,60]. Rationally, lower polyfunctionality and functional exhaustion of the T_eff_ cells with HAdV-C5 vaccine vectors would result in a reduced protection efficacy. However, differences in phenotypic T cell responses upon vaccination did not result in differences in protection in an infection challenge [17]. Therefore, additional factors may play a role. For example, a HAdV-C5 vaccine vector was shown to yield higher IFNγ and IL-2 but lower IL-4 producing cells compared to a recombinant Modified vaccinia Ankara (MVA) poxvirus vaccine [61]. This suggests of a more favorable Th1 bias and might compensate for lower polyfunctionality. Whether NHP-derived adenoviral vectors induce a Th1 response similar to HAdV-C5 has not been evaluated. In summary, it seems that there is no evidence that the cellular immune responses induced by NHP-derived adenoviruses are secondary to the responses induced by human-derived adenoviruses and, in fact, the former may be superior at the appropriate dose.

T_em_ cells are highly proliferative, polyfunctional cells with strong cytotoxic abilities and are therefore potential targets for anti-tumor immunity. Both HAdV-D26 and HAdV-B35, and several chimpanzee-derived adenoviral vectors were shown to induce higher expression of CD127, a marker for T_em_ cells, compared to HAdV-5 in mice [17,57]. Interestingly, vaccination of rhesus macaques with HAdV-D26 was shown to induce T_em_ cells which transitioned to a T_cm_ phenotype over time [62]. In contrast, antigen-specific T_em_ cells at mucosal sites did not transition to a T_cm_ phenotype, but rather remained a T_em_ phenotype which lasted over 2 years after vaccination with HAdV-D26 or HAdV-B35 [62,63]. Interestingly, NHP-derived adenoviral vectors seem to generate a T_cm_ phenotype faster. In human peripheral blood mononuclear cells (PBMC), the T cell response was more skewed towards a T_cm_ phenotype upon administration of ChAd3 already 4 weeks post-vaccination [59]. However, the increased memory formation observed in mice was lost at a lower dose, as the frequencies of memory T cells fell below the frequency observed with HAdV-C5 [50]. This phenomenon was not type-dependent, as it was observed for both ChAd3 and HAdV-E-classified ChAd63. These findings support the concept of rare serotypes to have lower immunogenicity at low doses and this might affect all CD8^+^ T cell subsets. With due considerations, it appears that NHP-derived adenoviral vectors are well able to generate long-term immunity and could therefore be preferable for use as oncolytic vectors.

Tissue-resident CD8^+^ memory T cells (T_rm_) have gained recent interest as potential effector cells in anti-tumor immunity. In a recent study, vaccination with MVA resulted in the recruitment of resident memory CD8^+^ T cells to APCs and led to the activation of the memory T cells, which preceded the recruitment of antigen-specific CD8^+^ T cell responses in a temporospatial manner [64]. In addition, introduction of viral peptides into a tumor was shown to activate residing virus-specific memory T cells surrounding the tumor tissue which lead to anti-tumor immunity [65]. The recruitment and activation of these resident memory T cells is referred to as the “bystander effect” and was shown to be CXCR3-dependent [64]. Interestingly, vaccination of mice with ChAd63, but not HAdV-C5, increased the production of CXCL10, a CXCR3-ligand [50]. This might provide ChAd63 with the means to induce stronger T_rm_ cell responses, similar to the T_cm_ cell responses mentioned above. Unexpectedly, a defect in the pathway had no effect on the generation of CD8^+^ T cell memory upon adenoviral infection [50]. This might be explained by the association of CD4^+^ T cell responses with virus-induced CXCL10 production, rather than CD8^+^ T cell responses [66]. Hence, a more detailed characterization of the specific CD4^+^ T cell responses induced by human- and NHP-derived adenoviruses might provide some insight in the generation of (anti-tumor) immune memory formation. In addition, a recent paper by Ahrends et al. showed that CD4^+^ T cells are essential for the maintenance of both T_em_ and T_cm_ cells [67]. Interestingly, ChAd3 induces CD4^+^ T cell responses more frequently than CD8^+^ T cell responses in humans [68]. However, no transcriptional differences in total CD4^+^ T cells after vaccination with human- and NHP-derived adenoviruses could be found in a NHP model [69]. Still, the use of bulk CD4^+^ T cells in the analyses has made it is possible for subtle differences within distinct subsets to go undetected.

One T cell subset that was shown to correlate with the protection of animals vaccinated with human- or NHP-derived adenoviruses were the CD4^+^ follicular B cell helper T cells (T_fh_) [69]. These lymph node-residing T_fh_ are associated with the development of functional B cell responses upon vaccination with an adenoviral vector [70]. Interestingly, B cells have immune-modulating effects that can be either pro-tumorigenic or anti-tumorigenic, although the mechanisms that underlie these are not yet fully understood. A recent study by Ehrenberg et al. performed a comparative analysis of two independent studies on an infection challenge in NHPs using HAdV-D26 as a vaccine carrier [71]. The data revealed an enriched gene signature in B cells that was associated with uninfected animals. Surprisingly, no such observations were made for the other lymphocyte subsets. As expected, the B cell signature was correlated with several antibody-mediated immune responses as well as IFNγ. Few studies have compared gene expression patterns induced by various adenoviral vectors. However, the expression of interferons in the dLN was shown to be differentially regulated between (non-)human primate-derived adenoviral vectors. Expression of IFNγ and IFNβ was shown to be substantially downregulated at earlier time points for HAdV-C5 and ChAd3, as compared to HAdV-D28, HAdV-B35, ChAd63, SAdV-11, and SAdV-16 [50]. It is noteworthy that the differential expression of interferons seems to be dependent on adenovirus species rather than host origin, as HAdV-C5 and ChAd3 both belong to the human species C adenoviruses while the others belong to the HAdV species B (HAdV-B35), species D (HAdV-D28), species E (ChAd63), species G (SAdV-11), and the SAdV species E (SAdV-16) viruses. Similar observations were made for wildtype adenoviruses, where the induction of genes involved in innate immunity was shown to differ between adenoviruses belonging to different species [72]. Nevertheless, the observation that the most potent adenoviral vectors (HAdV-C5 and ChAd3) are associated with reduced innate immune responses is puzzling.

In conclusion, it appears that the magnitude, endurance, and phenotype of the immune response is variable between adenovirus types. While there is great similarity between immune responses induced by (non-)human primate-derived adenoviruses, there are slight phenotypic differences in innate immunity responses and memory formation between rare serotypes and HAdV-C5. In accordance with earlier conclusions, the observed variability appears to be dependent on (sero)type rather than host origin.

## 6. Pathology in Humans

Adenoviruses are known respiratory and gastrointestinal pathogens in humans, as described in Lynch et al. [73]. However, in immune-competent individuals, adenoviral infections are mostly self-limited and often asymptomatic, or cause only mild flu-like symptoms. Most individuals encounter adenoviruses during childhood, as illustrated by a higher detection of adenoviral infections in young children [74]. In NHPs, adenoviruses also primarily infect infants and are not associated with severe disease [75,76]. Persistent adenoviral infection in a NHP species was shown to associate with an altered microbiome [77]. In addition, live adenoviruses are persistently shed from NHPs [78]. Therefore, adenoviruses seem to have a niche in the gastrointestinal tract of NHPs. Taken together, NHP-derived adenoviruses appear to be enzootic pathogens which cause only limited disease in their natural host, similar to humans-derived adenoviruses.

Adenoviruses have the potential to cross the species barrier, especially between human and NHPs. There are several reports of anthropozoonoses, the transmission of adenoviruses from NHP to human, but these are often based on phylogenetic or serological evidence and not associated with any disease [79]. As stated earlier, the frequency of cross-species transmission of (non)-human primate adenoviruses is thought to be very low [12]. This could be the result of (i) inefficient transmission due to incompatibility of the hosts, or (ii) inadequate detection of such transmissions. The retrieval of identical adenoviral DNA sequences from samples taken at several locations and in distinct NHP species speaks for the latter [80]. Moreover, the recent identification of HAdV-B76, the novel recombinant of human- and NHP-derived adenoviruses described earlier, further supports this hypothesis [20]. The isolation of HAdV-B76 from a patient with upper respiratory disease, which ultimately resulted in the untimely death of the patient, might raise concern of such recombinant viruses as new emerging pathogens. However, it should be noted that this isolated case did not result in any detectable outbreak. More importantly, there are still no reports of humans infected with wildtype NHP-derived adenoviruses despite the growing human-NHP interface [81]. The continuous discovery of novel adenovirus types that emerged from genetic recombination between human- and NHP-derived adenoviruses over the past decade hints towards a more frequent, but often undetected, cross-barrier mingling of adenovirus types. If this hypothesis were true, the pathogenicity of NHP-derived adenoviruses could possibly be overestimated. The accumulating number of clinical trials using NHP-derived adenoviral vaccine vectors verifies the safety of such replication-deficient vectors. At the time of writing, 11 clinical trials with ChAd3 have been completed which reported no severe adverse effects (https://clinicaltrials.gov/, accessed on 22 April 2020). Obviously, the use of replication-competent adenoviruses could possibly lead to additional side-effects. However, replication-defective viruses as well elicit immune responses to the vector. A phase I clinical study with ChAdOx2, an *E1/E3*-deleted vector based on ChAd68 in which the *E4 ORF6/7* region is replaced by the equivalent region of HAdV-C5, showed the seroconversion of seronegative individuals upon administration of the vaccine [82]. The generation of NAbs to the vector exemplifies the recognition of the virus as an invading pathogen by the immune system. Therefore, it seems likely that replicating NHP-derived adenoviruses are not regarded as novel pathogens by the immune system and will in essence not give rise to a disproportionate immune response. Conversely, the clinical application of NHP-derived adenoviruses in cancer patients could yield some unforeseen complications that might affect the efficacy of the vector. For example, the level of NAbs against ChAd68, but not HAdV-C5, was slightly increased in cancer patients as opposed to healthy controls and was dependent on tumor type and clinical stage [83]. The influence of these NAbs on the efficacy of adenoviral therapy is still under debate. For reovirus, another candidate virus for oncolytic virotherapy, it was shown how neutralizing antibodies can lead to improved anti-tumor responses due to the monocyte-mediated hand-off of infectious virus to the tumor [84]. Similar observations could be made for coxsackievirus but not herpes simplex virus, which illustrates the variability between viruses. Both pre-clinical and clinical studies have shown widely contradicting results regarding the effect of neutralizing antibodies on adenoviral therapy [85]. Whereas some studies showed that pre-existing immunity leads to an aggravated immune response which results in higher toxicity and mortality, other studies demonstrated no effect of NAbs or even improved treatment outcomes. The safety observed in clinical trials using NHP-derived adenoviral vectors could speak in favor of the disadvantageous effects of neutralizing antibodies. Meanwhile the controversial effects of pre-existing neutralizing immunity are being elucidated, the use of low-prevalent adenoviruses could provide a means to circumvent the issue. The low prevalence of NAbs directed against NHP-derived adenoviruses creates a more homogenous baseline immunity in the patient population. This may reduce the response variability currently observed in clinical trials using HAdV-C5-based vectors. In this way, smaller patient cohorts could suffice to demonstrate efficacy of treatment. In summary, there is as of yet no substantial evidence that disputes the safe use of NHP-derived adenoviral vectors as oncolytic agents in humans. Moreover, their low-seroprevalence provides more uniformity in antiviral immunity in patients, which could increase the power of small-group clinical trials.

## 7. Future Perspectives

NHP-derived adenoviruses have formed the basis for several promising vaccine vectors due to their similarity to human-derived adenoviruses and their low seroprevalence in the human population. Following their initial success, the use of such non-human adenoviral vectors could be broadened to the use as oncolytic agents. In light of the current pandemic, it is anticipated that the use of non-human vectors in humans is met with uneasiness. Therefore, this review aimed to summarize the present knowledge on human- and NHP-derived adenoviruses, with a focus on their potential use as oncolytic agents. Current virus taxonomy classifies adenoviruses according to the host they originate from. However, the genesis of adenoviruses via recombination of viruses derived from distinct host species hampers the current taxonomic labeling of human- and NHP-derived adenoviruses. For NWMs, a common name was given to adenoviruses which were able to infect multiple species [80]. Given their shared ancestry, recurrent and bidirectional cross-species transmission, and their unrestrained ability for recombination, a unified labeling might be applicable to the human- and NHP-derived adenoviruses as well, as has been suggested previously [86]. Variations in receptor use, replication potential, and oncolytic abilities appear to be characteristic for adenovirus (sero)types rather than their host origin. Furthermore, there is great similarity between immune responses induced by (non-)human primate-derived adenoviruses; albeit, there are slight phenotypic differences in innate immunity responses and immune memory formation between rare serotypes, either human or NHP-derived, and HAdV-C5. The induction of polyfunctional T_eff_ cells and higher counts of T_em_ and T_cm_ cells by NHP-derived adenoviral vectors could possibly result in improved anti-tumor immunity. Therefore, oncolytic vectors derived from NHP-derived adenoviruses might prove to generate more durable responses. In addition, the sporadic pathogenicity of NHP-derived adenoviruses in humans and the safety assessments of replication-deficient NHP-derived adenoviral vectors in a number of clinical trials provides consolation regarding the use of replication-competent viruses. Nevertheless, data on replication-competent NHP-derived adenoviruses in humans are so far insufficient and leave some characteristics, including oncolytic ability and immune evasion, largely unexplored. Given the evidence in this review, the use of NHP-derived adenoviruses could be considered equally suitable in respect to the use of human-derived adenoviruses as the bedrock for the development of new oncolytic adenoviral vectors (Figure 3). Nonetheless, bearing in mind that the development of oncolytic vectors from NHP-derived adenoviruses is yet at its infancy, there is still a road ahead in delineating the true oncolytic potential of these viruses.

## Figures and Tables

**Figure 1 ijms-21-04821-f001:**
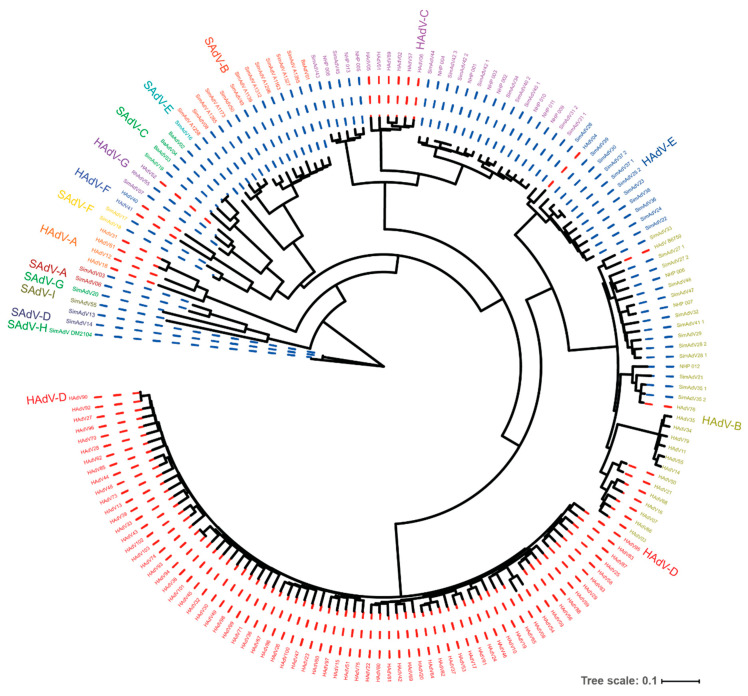
Genetic resemblance of human- and non-human primate (NHP)-derived adenoviruses. A phylogenetic tree was constructed based on whole genome sequences of the adenoviruses that could be retrieved from Genbank. The *Human* and *Simian Adenovirus* species are color coded, and the colored dotted line indicates the adenovirus was isolated from either a human host (red) or NHP host (blue). The tree makes evident that the *Human Adenovirus* (*HAdV*) species HAdV-B, C, E, and G contain types isolated from human as well as from NHPs, suggesting cross-species transfer. The large group of HAdV-D types all originate from humans. The isolates numbered NHP-001 through NHP-013 represent new isolates from NHPs held in captivity (unpublished data). To generate the tree, sequences were assembled in FASTA format, aligned with MAFFT, and curated using BMGE. From the aligned data, a tree was generated using FastME. The process was performed at the NGPhylogeny.fr server by means of the “Oneclick workflow” using default parameters. The tree was exported to the Interactive Tree of Life server and visualized in circular mode. A high-resolution version of the figure is provided as Appendix A.

**Figure 2 ijms-21-04821-f002:**
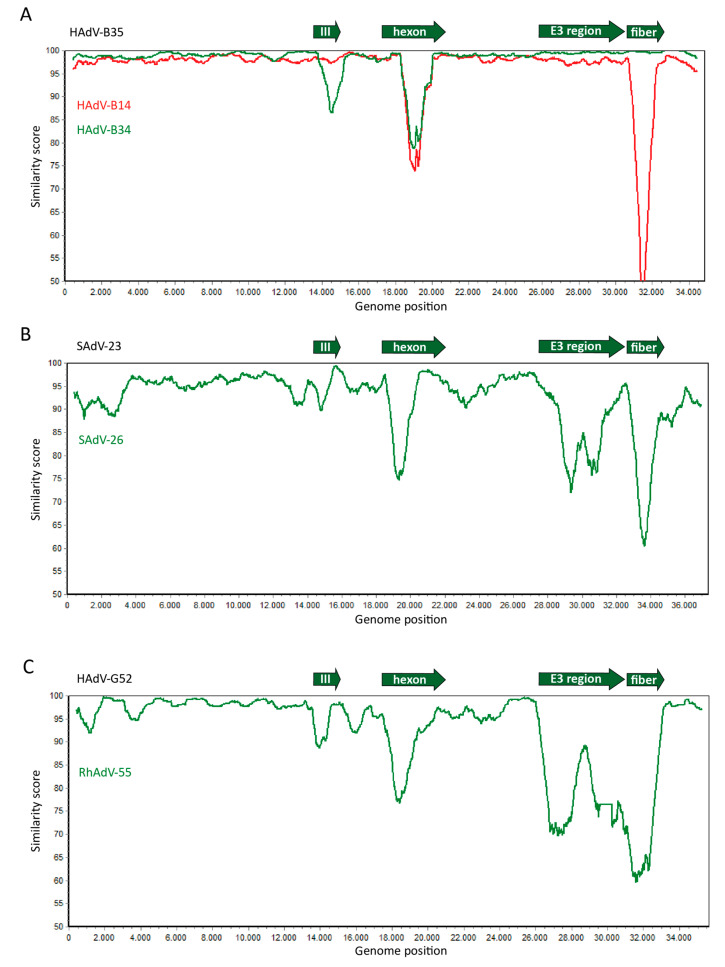
Similar genome sequence diversity between human- and non-human primate (NHP)-derived adenoviruses. Whole genome sequences of human- and NHP-derived adenoviruses were compared and the similarity between the sequences was plotted in Simplot. (**A**) The sequences of *Human Adenoviruses* (HAdV)-B14 (red line) and HAdV-B34 (green line) were aligned with HAdV-B35. (**B**) The sequences of *Simian Adenoviruses* (SAdV)-23 and SAdV-26, both chimpanzee-derived adenoviruses and members of the HAdV-E species, were aligned. (**C**) The sequences of HAdV-G52 and rhesus monkey-derived RhAdV-55 of species HAdV-G were aligned. These data demonstrate that in human adenoviruses, as well as in the adenoviruses isolated from NHPs, the main sequence diversity is found in the genes encoding the major capsid proteins penton-base (protein III), hexon, and fiber, and the genes located in the *E3* region. This is consistent with the relatively frequent exchange of these gene segments between different adenoviruses. The plots were generated by aligning the nucleotide sequences of the human and simian adenoviruses as retrieved from Genbank according to the Smith–Waterman algorithm [29] using default parameters in case of two sequences, or by MAFFT V7.407 [30] using default parameters for multiple sequences. The sequence similarity graphs were generated using Simplot [31] using a window size 800, and step size 10.

**Figure 3 ijms-21-04821-f003:**
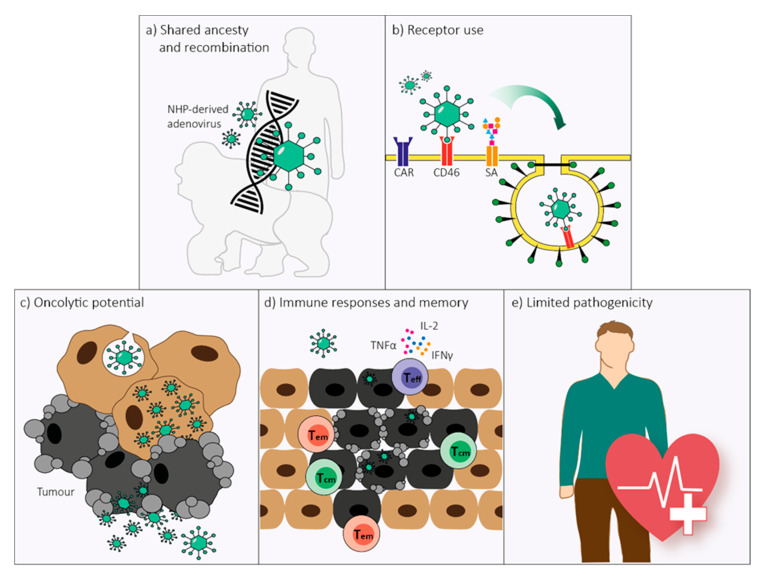
Non-human primate (NHP)-derived adenoviruses possess five main characteristics that would make them equally suitable as human-derived adenoviruses for use as oncolytic agents. (**a**) Human- and NHP-derived adenoviruses have shared ancestry and demonstrated recurrent and bidirectional cross-species transmission over the course of evolution. The homologous genomic organization of human- and NHP-derived adenoviruses allows for unrestrained recombination which transcends host origin. (**b**) Several attachment receptors are conserved between human- and NHP-derived adenoviruses, including the coxsackie and adenovirus receptor (CAR) and CD46, and possibly sialic acids (SA). Receptor-specificity depends on human adenovirus species rather than host origin. (**c**) NHP-derived adenoviruses can have oncolytic abilities in human tumor cells. Susceptibility to oncolysis is adenovirus type-dependent, similar to what is observed for human-derived adenoviruses. (**d**) The magnitude, endurance, and phenotype of the immune response elicited by NHP-derived adenoviruses are not substantially different from the responses elicited by human-derived adenoviruses. As NHP-derived adenoviral vectors induce higher frequencies of polyfunctional T effector (T_eff_) cells, as well as effector memory (T_em_) and central memory (T_cm_) T cells, they could be preferable for use as oncolytic vectors. (**e**) Similar to human-derived adenoviruses, NHP-derived adenoviruses cause only limited disease in their natural host. The safety of replication-deficient NHP-derived adenoviral vectors has been supported by multiple clinical studies and provides a basis for the safe use of NHP-derived adenoviral vectors as oncolytic agents in humans.

**Table 1 ijms-21-04821-t001:** Nomenclature of adenoviruses.

Nomenclature of Human and Non-Human Primate (NHP)-Derived Adenoviruses
The adenoviruses isolated from humans and NHPs are all classified in the *Mastadenovirus* genus. Within this genus, the *Human Adenoviruses* (HAdV) are grouped in “species” (formerly “subgroups”) *A* through *G*. Similarly, NHP adenoviruses are grouped in *Simian Adenovirus* (SAdV) *A* through *I*. Within these species the isolates are clustered in “serotypes” or “types”. To date, 103 types are distinguished in adenoviruses isolated from humans. The naming of adenoviruses reflects the host from which the first type of an adenovirus species has been isolated, the adenovirus species, and their type. The *Human Adenovirus* species C type 5 is indicated as HAdV-C5.

**Table 2 ijms-21-04821-t002:** NHP-derived adenoviruses for use as vaccine vectors discussed in this review.

Adenovirus ^1^	Short Names ^2^	Host Origin	Species	Genbank Accession No.
Chimpanzee adenovirus Y25	ChAdOx1	Chimpanzee	HAdV-E	JN254802
Gorilla beringei adenovirus 7	GC44	Gorilla	HAdV-B	KC702813 ^3^
Gorilla beringei adenovirus 8	GC45	Gorilla	HAdV-B	KC702814 ^3^
Gorilla beringei adenovirus 9	GC46	Gorilla	HAdV-B	KC702815 ^3^
Simian adenovirus 11	SAdV-11	Rhesus Macaque	HAdV-G	KP329562
Simian adenovirus 16	SAdV-16	Grivet	SAdV-E	KP329564
Simian adenovirus 32	SAdV-B32	Chimpanzee	HAdV-B	FJ025911
Simian adenovirus 21	AdC1	Chimpanzee	HAdV-B	AC_000010
Simian adenovirus 23	AdC6	Chimpanzee	HAdV-E	AY530877
Simian adenovirus 24	AdC7	Chimpanzee	HAdV-E	AY530878
Simian adenovirus type 25.2	ChAd68/ChAdOx2	Chimpanzee	HAdV-E	FJ025918.1
Pan paniscus adenovirus type 3	PanAd3	Bonobo	HAdV-C	N.A.
Pan troglodytes adenovirus type 3	ChAd3	Chimpanzee	HAdV-C	CS479276
Pan troglodytes adenovirus type 63	ChAd63	Chimpanzee	HAdV-E	CS479277

^1^ The naming of non-human adenoviruses is not standardized as is customary for human adenoviruses. We refer here to these viruses by the names commonly used in literature. ^2^ Generic short names for the adenoviruses commonly used in literature, ^3^ Contains the hexon sequence only.

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
