# Peer review of "Non-Human Primate-Derived Adenoviruses for Future Use as Oncolytic Agents?"

_ijms, 2020, doi:10.3390/ijms21144821_

Round 1
Reviewer 1 Report
The manuscript constitutes a very interesting aspect to review non-human primate (NHP)-derived adenoviruses, regarding the potential to apply as oncolytic agents. It not only covers the previous data but also analyzes the gap and point out the essential future perspective. I enjoyed reading the review, and have only some minor comment:
1. Regarding the title, would ‘Non-Human Primate-Derived Adenoviruses for Use as Oncolytic Agents?’ more straightforward? For what has been discussed in the manuscript is not the race or to choose between human and non-human Ads, rather offer a new path for oncolytic Ads development using non-human Ads.
2. There are two ‘table 1’ in line 63 and 117. Seems like the table 1 in line 63 is what in text line 44-mentioned textbox 1.
3. The figure 1 offers important information. However, in the print version, it is not readable, the ad type labels are too small.
4. Should the figure 2 position earlier? It is first cited in ‘2. Genome Organization and Recombination’
5. In figure 2c, it should be HAdV-G52 not G55.
Author Response
The manuscript constitutes a very interesting aspect to review non-human primate (NHP)-derived adenoviruses, regarding the potential to apply as oncolytic agents. It not only covers the previous data but also analyzes the gap and point out the essential future perspective. I enjoyed reading the review, and have only some minor comment:
- Regarding the title, would ‘Non-Human Primate-Derived Adenoviruses for Use as Oncolytic Agents?’ more straightforward? For what has been discussed in the manuscript is not the race or to choose between human and non-human Ads, rather offer a new path for oncolytic Ads development using non-human Ads.
We amended the title as suggested. We agree with your statement that the manuscript discusses the use of non-human primate-derived adenoviruses alongside the use of human adenoviruses for their use as oncolytic agents. However, the manuscript also discusses several characteristics of non-human primate (NHP)-derived adenoviruses (no pre-existing immunity, the generation of a more pronounced immune memory phenotype) which would make them more favorable compared to human adenoviruses. On the other hand, their replication potential and oncolytic abilities have only moderately been addressed so far. This is fairly summarized in the discussion (final sentence, line 612): “Nonetheless, bearing in mind that the development of oncolytic vectors from NHP-derived adenoviruses is yet at its infancy, there is still a road ahead in delineating the true oncolytic potential of these viruses.” The lack of data on some of these key characteristics makes it impossible to determine whether non-human primate-derived adenoviruses can provide an (better) alternative for the use of human adenoviruses as oncolytic agents. Therefore, we believe the question still remains whether human or non-human primate-derived adenoviruses are more preferable as oncolytic agents, or whether they are equally suitable.
There are two ‘table 1’ in line 63 and 117. Seems like the table 1 in line 63 is what in text line 44-mentioned textbox 1.
We corrected this error.
The figure 1 offers important information. However, in the print version, it is not readable, the ad type labels are too small.
We included a high resolution version of the figure as Supplementary Figure S1.
Should the figure 2 position earlier? It is first cited in ‘2. Genome Organization and Recombination’
We replaced the figure as suggested.
In figure 2c, it should be HAdV-G52 not G55.
Thank you for pointing out this error. We corrected it.
Reviewer 2 Report
The Review submitted by Bots et al is about the use of NHP-derived Ads as oncolytic agents
1) Appropriated references about oncolytic viruses are missing especially concerning their use as vaccine strategy for immunogenic tumors such as melanoma (Capasso C, Magarkar A, Oncoimmunology 2017)
2)Authors should also report the different route of administrations and pro and cons related to the administration of viruses especially considering future translation in patients
Author Response
The Review submitted by Bots et al is about the use of NHP-derived Ads as oncolytic agents
1) Appropriated references about oncolytic viruses are missing especially concerning their use as vaccine strategy for immunogenic tumors such as melanoma (Capasso C, Magarkar A, Oncoimmunology 2017)
We agree that the use of oncolytic adenoviruses as cancer vaccines is an interesting topic and a valuable approach to target immunogenic tumors. The reference that you suggested describes an automated algorithm-driven platform for the discovery of tumor antigens and does not seem to relate to the topic of the manuscript. In the article however, a reference is made to Capasso et al. (2015) which describes the use of human conditionally replicating adenoviruses (CrAds) which are bound to genetically modified positively charged MHC-I-restricted peptides (PeptiCrads). This approach seems promising and it would be worthwhile to explore this approach with non-human primate (NHP)-derived CrAds. We cited the latter reference in our manuscript (line 115).
2) Authors should also report the different route of administrations and pro and cons related to the administration of viruses especially considering future translation in patients
We have included a brief discussion on the different routes of administration as they are an important aspect for the clinical translation of virotherapy. Intravenous administration of oncolytic viruses would be most preferable as the treatment is first of all least invasive. In addition, secondary tumors can go undetected, and, more importantly, not all tumors are accessible by needle. To highlight this, we have incorporated a small notion on the issue in the manuscript (line 75-77). Intravenous administration would probably benefit most from the use of NHP-derived adenoviruses as the absence of neutralizing antibodies will allow the virus to travel through the bloodstream and to the tumor, as was already stated in the manuscript in line 73. The manuscript shows that there appear to be minor biological differences between human and NHP-derived adenoviruses regarding their genome structure, receptor use, replication and oncolysis, immune modulation, and pathogenicity. Consequently, we expect that there will only be minor differences between human and NHP-derived adenoviruses regarding the different routes of administration. In addition, there is as of yet no data on the different routes of administration of oncolytic NHP-derived adenoviruses. Therefore, we are not able to discuss the topic any further in the manuscript.
Reviewer 3 Report
Bots and Hoeben review similarities and differences between human and non-human primate adenoviruses and, on the basis of this comparison, propose that NHP adenoviruses could perhaps be used to derive more effective oncolytic agents for the treatment of cancer in humans.
This is a well written, comprehensive manuscript on an interesting subject. The major concern I have with this work is that it cannot be considered a review on the subject of NHP-derived adenoviruses as oncolytic agents (as the title and abstract suggest), simply because this is a yet virtually nonexistent field. Thus far, these viruses have been used as vaccines, but not as oncolytic agents. Consequently, apart from unpublished observations made by the authors, only a single published laboratory study on this subject is cited and discussed. Rather, the authors extensively review the molecular and functional differences between human and non-human primate adenoviruses; in view of their hypothesis that NHP adenoviruses could perhaps be used as oncolytic agents. In this regard, if IJMS allows this format, the work should better be considered an opinion/perspective piece. Alternatively, the manuscript could be presented as a review on the differences and similarities between human and NHP adenoviruses, which could culminate into a provoking future perspective on the possible use of NHP adenoviruses as oncolytic agents. This requires some rephrasing of title and abstract in particular.
Further, the use of NHP-derived viruses is proposed as a strategy to avoid clearance by neutralizing antibodies. To judge the importance of this aspect, the manuscript could do with a short review of the known influence of neutralizing antibodies in humans on the treatment efficacy of oncolytic immunotherapy using human adenoviruses.
Author Response
Bots and Hoeben review similarities and differences between human and non-human primate adenoviruses and, on the basis of this comparison, propose that NHP adenoviruses could perhaps be used to derive more effective oncolytic agents for the treatment of cancer in humans.
This is a well written, comprehensive manuscript on an interesting subject. The major concern I have with this work is that it cannot be considered a review on the subject of NHP-derived adenoviruses as oncolytic agents (as the title and abstract suggest), simply because this is a yet virtually nonexistent field. Thus far, these viruses have been used as vaccines, but not as oncolytic agents. Consequently, apart from unpublished observations made by the authors, only a single published laboratory study on this subject is cited and discussed. Rather, the authors extensively review the molecular and functional differences between human and non-human primate adenoviruses; in view of their hypothesis that NHP adenoviruses could perhaps be used as oncolytic agents. In this regard, if IJMS allows this format, the work should better be considered an opinion/perspective piece. Alternatively, the manuscript could be presented as a review on the differences and similarities between human and NHP adenoviruses, which could culminate into a provoking future perspective on the possible use of NHP adenoviruses as oncolytic agents. This requires some rephrasing of title and abstract in particular.
Thank you kindly for your feedback. You have pointed out a relevant concern regarding the format of the review. Apart from the article by Cheng et al. (2017) and our own soon to be published data, there are indeed no studies on the potential of non-human primate (NHP)-derived adenoviruses as oncolytic agents. In view of this, we would agree with the work could be viewed as an opinion/perspective paper. In order to assure that the reader is not misled by either the title or the abstract, we have adjusted both the title (line 2-3) and the abstract (line 20, 21, and 24) to stress the intent of our manuscript.
Further, the use of NHP-derived viruses is proposed as a strategy to avoid clearance by neutralizing antibodies. To judge the importance of this aspect, the manuscript could do with a short review of the known influence of neutralizing antibodies in humans on the treatment efficacy of oncolytic immunotherapy using human adenoviruses.
The impact of neutralizing antibodies on the efficacy of oncolytic immunotherapy is indeed important as the effects have shown to be quite counterintuitive sometimes. For example, Berkeley et al. (2018) demonstrated for reovirus how neutralizing antibodies can lead to improved anti-tumor responses due to the monocyte-mediated hand-off of infectious reovirus to the tumor. Similar observations could be made for Coxsackievirus but not HSV, which illustrates the variability between viruses. For adenoviruses, there is no consensus yet but the presence of neutralizing antibodies is generally believed to be disadvantageous. In response to your comments, we have made serval adjustments to the manuscript:
1) Line 79 We have briefly mentioned the implications of pre-existing immunity (i.e. neutralizing antibodies) on the efficacy of human adenovirus treatments to support the interest in low-seroprevalent adenoviruses, like NHP-derived adenoviruses for use as oncolytic agents.
2) Line 542-562 We have included a paragraph on the influence of neutralizing antibodies on the efficacy of adenovirus therapy, which has been nicely described in Zaiss et al. (2009) ‘The influence of innate and pre-existing immunity on adenovirus therapy’. While there is no consensus regarding the effects of neutralizing antibodies, the use of NHP-derived adenoviral vectors will provide a more homogenous baseline immunity in patients which could lower patient-to-patient variation. Moreover, reduced variability will benefit the outcome of clinical trials regardless of the effects of neutralizing antibodies on oncolytic adenoviral therapy. Therefore, the use of NHP-derived adenoviruses could provide a means to improve the clinical assessment of oncolytic adenovirus therapy.
Round 2
Reviewer 2 Report
I suggest the MS for publication